# Enduring, Sexually Dimorphic Impact of In Utero Exposure to Elevated Levels of Glucocorticoids on Midbrain Dopaminergic Populations

**DOI:** 10.3390/brainsci7010005

**Published:** 2016-12-30

**Authors:** Glenda E. Gillies, Kanwar Virdee, Ilse Pienaar, Felwah Al-Zaid, Jeffrey W. Dalley

**Affiliations:** 1Division of Brain Sciences, Imperial College London, Hammersmith Hospital, Du Cane Road, London W12 0NN, UK; i.pienaar@imperial.ac.uk (I.P.); f.al-zaid14@imperial.ac.uk (F.A.-Z.); 2Department of Psychology, University of Cambridge, Downing Street, Cambridge CB2 3EB, UK; kv10001@cam.ac.uk (K.V.); jwd20@cam.ac.uk (J.W.D.); 3Behavioural and Clinical Neuroscience Institute, University of Cambridge, Downing Street, Cambridge CB2 3EB, UK; 4Department of Applied Sciences, Faculty of Health and Life Sciences, Northumbria University, Newcastle Upon Tyne NE1 8ST, UK; 5Department of Psychiatry, University of Cambridge, Cambridge CB2 2QQ, UK

**Keywords:** dopaminergic neurones, astrocytes, midbrain, ventral tegmental area, substantia nigra pars compacta, antenatal glucocorticoid treatment, developmental programming, sex dimorphisms

## Abstract

Glucocorticoid hormones (GCs) released from the fetal/maternal glands during late gestation are required for normal development of mammalian organs and tissues. Accordingly, synthetic glucocorticoids have proven to be invaluable in perinatal medicine where they are widely used to accelerate fetal lung maturation when there is risk of pre-term birth and to promote infant survival. However, clinical and pre-clinical studies have demonstrated that inappropriate exposure of the developing brain to elevated levels of GCs, either as a result of clinical over-use or after stress-induced activation of the fetal/maternal adrenal cortex, is linked with significant effects on brain structure, neurological function and behaviour in later life. In order to understand the underlying neural processes, particular interest has focused on the midbrain dopaminergic systems, which are critical regulators of normal adaptive behaviours, cognitive and sensorimotor functions. Specifically, using a rodent model of GC exposure in late gestation (approximating human brain development at late second/early third trimester), we demonstrated enduring effects on the shape and volume of the ventral tegmental area (VTA) and substantia nigra pars compacta (SNc) (origins of the mesocorticolimbic and nigrostriatal dopaminergic pathways) on the topographical organisation and size of the dopaminergic neuronal populations and astrocytes within these nuclei and on target innervation density and neurochemical markers of dopaminergic transmission (receptors, transporters, basal and amphetamine-stimulated dopamine release at striatal and prefrontal cortical sites) that impact on the adult brain. The effects of antenatal GC treatment (AGT) were both profound and sexually-dimorphic, not only in terms of quantitative change but also qualitatively, with several parameters affected in the opposite direction in males and females. Although such substantial neurobiological changes might presage marked behavioural effects, in utero GC exposure had only a modest or no effect, depending on sex, on a range of conditioned and unconditioned behaviours known to depend on midbrain dopaminergic transmission. Collectively, these findings suggest that apparent behavioural normality in certain tests, but not others, arises from AGT-induced adaptations or compensatory mechanisms within the midbrain dopaminergic systems, which preserve some, but not all functions. Furthermore, the capacities for molecular adaptations to early environmental challenge are different, even opponent, in males and females, which may account for their differential resilience or failure to perform adequately in behavioural tests. Behavioural “normality” is thus achieved by the midbrain dopaminergic network operating outside its normal limits (in a state of allostasis), rendering it at greater risk to malfunction when challenged in later life. Sex-specific neurobiological programming of midbrain dopaminergic systems may, therefore, have psychopathological relevance for the sex bias commonly found in brain disorders associated with these systems, and which have a neurodevelopmental component, including schizophrenia, ADHD (attention/deficit hyperactivity disorders), autism, depression and substance abuse.

## 1. Introduction

Mental health problems are widely recognised as leading causes of illness and disability. The World Health Organisation (WHO) reported that worldwide one in four adults will be affected by mental and behavioural disorders during their lives. Mental health issues are also on the rise, especially in children and young adults, in whom various types of mental illness have doubled over the last decade, such that now one in five young people under the age of 18 are affected with conditions that may remain with them into adulthood. In addition to improving treatments, the most recent emphasis by global bodies, such as WHO, the United Nations, charities and governments, has been on removing the stigma and discrimination attached to mental disorders, as well as addressing the reluctance of those who suffer from these conditions to come forward for diagnosis and treatment. However, scientific advances are also being made to understand the origins of mental illness, which will be key for developing better therapeutic strategies and for realising the potential to reverse or prevent these conditions. In addition to genetic make-up, we now know that environmental and social factors and stressors interact to influence an individual’s susceptibility to becoming mentally ill [1,2,3]. We also understand that mental illness is associated with changes in the structure, chemistry and function of the brain; moreover, these changes are superimposed on innate differences in brain anatomy, neurochemistry and function according to one’s sex, as defined by reproductive organs (testes or ovaries) and sex chromosomes (XX in males, XY in females) [4,5]. These physiological and structural brain differences are thought to underpin sex differences that are manifest in virtually all aspects of neurological and psychological disorders, ranging from susceptibility, prevalence, presentation and symptoms to disease progression, pathology and response to treatments. Collectively, these observations lead to the conclusion that there is, indeed, a biological basis for mental health conditions, as well as for the sex bias which characterises them, indicating that diagnostic, treatment and preventative strategies should be differentially targeted in men and women. Therefore, we urgently need to understand how changes in the brain that lead to brain disorders are created.

A large body of evidence from human and animal research has established that perturbations in the normal environment during early life, especially fetal life, programme changes in brain structure and function and profoundly affect vulnerability to disease, including mental health problems, that may emerge in childhood, adolescence or adulthood [3,6,7,8,9]. However, the precise neural substrate and the mechanisms which link the early environment to the development of adult psychopathology remain poorly characterised. In this review we shall focus on the midbrain dopaminergic systems as the neural substrate and on exposure during prenatal life to inappropriately elevated levels of glucocorticoids (GCs) as a mechanism which disrupts the normal developmental trajectories of these systems. After summarising briefly the roles of the midbrain dopaminergic systems in health and disease, we shall consider the evidence that these systems are highly sensitive to disturbances in the intrauterine environment by a broad range of stressors, and that GCs, commonly released in response to all types of stressors, are key factors responsible for early life programming in general, which differentially affect male and female brains. We shall then focus on how the structure, neurochemistry and function of the adult dopaminergic systems are altered in rodent male and female brains after brief antenatal treatment with GCs. Elucidating links between developmental exposure to GCs and dopaminergic malfunction in adulthood could, therefore, contribute to our understanding of the aetiology of brain disorders which, in most instances, differentially affect men and women. This may also have importance from a clinical perspective in view of the association between childhood behavioural deficits with repeated courses of synthetic GCs that are commonly prescribed to accelerate fetal lung function in the management of women at risk of preterm delivery [10,11,12].

## 2. Midbrain Dopaminergic Systems

### 2.1. Function

The midbrain dopaminergic systems are critical for integrating mammalian behavioural responses and adaptations to changes in the environment, thus enabling the individual to cope with what is perceived as a threat or stress [13,14,15,16]. Sub-sets of neurons in the ventral tegmental area (VTA) give rise to two major pathways: the mesolimbic dopaminergic (MLDA) pathway that innervates the ventral striatum, especially the nucleus accumbens (NAc), to regulate the process controlling emotion, reward and feeding, and the mesocortical dopaminergic (MCDA) pathway projecting to the prefrontal cortex, involved in cognition. Neurons in the substantia nigra pars compacta (SNc) form the nigrostriatal dopaminergic (NSDA) pathway innervating the dorsal striatum, which is central to sensorimotor integration, and may also interact with the MLDA under certain circumstances [17]. These systems form a highly responsive, stressor–sensitive circuitry. Stress or aversive events activate sub-sets of VTA neurons to stimulate the process of learning and memory [18,19,20,21,22,23,24,25,26], thereby enabling the storage and recall of appropriate coping behaviours when similar situations are encountered subsequently. Stress-induced activation of the SNc neurons [20] contributes to altered locomotor activity that typically accompanies stress, and their activation may ultimately also feed into the cognitive systems [17,27].

### 2.2. Malfunction, Neurobiological Programming and Sex Dimorphisms

#### 2.2.1. Malfunction

While the majority of individuals, human or non-human, can cope well with stress, we know that others cannot, and this failure of coping mechanisms is an established risk factor for developing brain disorders [1,28,29,30]. Not surprisingly, therefore, malfunction of dopamine (DA)-dependent stress-coping circuitry is a prevalent factor in psychopathology, contributing to disorders such as schizophrenia [31], anxiety and depression [32], addictive behaviours [33,34] and attention/deficit hyperactivity disorders (ADHD) [35,36] that primarily involve the VTA DA systems. To some extent, this can be modelled experimentally, where exposure of rats to chronic stress also has consequences for the VTA system. Unlike the positive coping response to acute stress in the MLDA/MCDA pathways, tonic activity may be down-regulated after chronic stress [37], leading to reduced dopaminergic transmission to the medial prefrontal cortex (mPFC) and deficiencies in working memory [38,39]. This also has the consequence of increased DA release in the NAc, which is under inhibitory control from the mesocortical system, leading to behavioural change [40]. Stress dysfunction involving the SNc may further have a role in the aetiology of the pre-clinical, non-motor symptoms of Parkinson’s disease (PD, such as depression), as well as the later progression of the motor symptoms, which characterise the condition [41,42].

#### 2.2.2. Neurobiological Programming

Although the aetiologies of the aforementioned conditions associated with dopaminergic malfunction remain poorly understood, it is now well established that during critical stages of development, especially fetal life, stress due to a host of different causes, can have detrimental effects on the maturing brain, with implications for abnormal behaviours, emotional problems and the emergence of neurological and neuropsychiatric conditions in later life [3,7,8,9,31,43,44,45]. Moreover, the midbrain dopaminergic systems appear to be central to this phenomenon. For example, it has been estimated that, in addition to a genetic neurodevelopmental component, 20%–30% causation in schizophrenia may be non-genetic and attributable to an adverse or sub-optimal intra-uterine environment arising from stressors, including obstetric complications, exposure of the mother to natural disasters (such as famine and malnutrition), death of a loved one, low socio-economic status and infections (such as influenza) [46,47,48,49,50]. Perinatal/obstetric complications and prenatal stressors are also associated with ADHD [51] and autism spectrum disorders [52]. Additionally, low birth weight (typically taken as a marker of intra-uterine metabolic stress) and other stressors have been associated with an increased risk for developing anxiety and depression [2,53,54]. An adverse intra-uterine environment may also contribute to causing or predisposing the NSDA system to degenerate in PD [55]. Although such epidemiological evidence is largely associative and not proof of cause and effect, animal models provide convincing evidence in support of the view that the ascending dopaminergic systems are direct targets for neurobiological programming by intrauterine and perinatal stress. These include maternal exposure to restraint (psychogenic stress), hypoxia (modelling obstetric complications), immune challenge and malnutrition (low birth weight), which alter dopaminergic activity and DA-dependent functions in the adult offspring, such as DA release, DA transporter levels, DA-dependent learning, locomotor activity and addictive behaviours [40,46,56,57,58,59,60].

Collectively, both clinical and experimental observations support the view that central developing dopaminergic systems are highly sensitive to disruption by virtually any form of adverse environment in utero. Prominent among the mechanisms which may be responsible for early life programming in the brain is exposure to elevated levels of glucocorticoids (GCs). During late gestation there is a natural rise in the bioavailability of endogenous GC hormones (cortisol in some species such as humans, pigs and guinea-pigs; corticosterone in other species, including rats and mice) produced by the foetal/maternal adrenal cortex. This occurs at a critical developmental window and plays a fundamental role in the development of mammalian tissues and organs, including surfactant production in the lungs, essential for survival at birth, as well as processes of synaptogenesis and cell survival in the brain [61,62]. However, exposure to these ‘normal’ late gestational elevations of GCs outside the pre-term critical window, due to excessive/prolonged stress or therapeutic use, has proven to be deleterious, particularly in the brain. Hence, high maternal cortisol levels in pregnancy are associated with low birth weight, sensitization of the infant hypothalamo-pituitary-adrenal (HPA) axis response to stress, childhood behavioural disorders and altered brain structure [44,63,64,65,66,67,68]. Investigations in animal models amply support a mechanistic link between exposure of the developing foetus to inappropriate elevations in GCs and programming of an altered behavioural phenotype and changes in stress-responsive and metabolic profiles, with remarkable similarities to the clinical profiles [64,65,69,70,71,72]. The prominent sensitivity of dopaminergic populations to developmental programming underscores the need to understand how intrauterine elevations of GCs impact the developmental trajectories of these systems (Section 3).

#### 2.2.3. Sex Dimorphisms

The majority of animal studies that provide evidence for neurobiological programming of the dopaminergic systems have been performed only in males. This largely ignores the growing awareness of innate sex dimorphisms within the SNc and VTA systems themselves, which are reviewed elsewhere [73,74] as well as the extensive literature evidencing the qualitative as well as quantitative nature of sex differences in the impact of early life stressors and GCs on brain structure and function in both humans [3,9,75,76,77,78] and experimental species [74,78,79,80,81,82,83,84,85]. Clearly, this phenomenon is likely to have considerable translational significance for brain disorders, which show a sex difference in many respects. For example, DA-associated disorders with evidence of early-life risk factors often show a greater prevalence in males, ranging from several-fold (autism, ADHD [35,86]) to two-fold (PD [73]). Schizophrenia and PD also have a significantly earlier onset in males and show sex differences in presentation, symptoms and progression [73,74,87]. Sex differences are also prevalent in all phases of drug abuse in humans [88,89,90] and pre-clinical studies identify prenatal stress as a sex-specific risk factor for different aspects of substance abuse [84]. It is, therefore, essential that we consider the impact of intrauterine elevations of GCs on the midbrain dopaminergic systems in both males and females.

## 3. Glucocorticoids and Programming of Midbrain Dopaminergic Systems

Early studies of the impact of intrauterine stress or inappropriate elevations in GCs focused on the endocrine arm of the stress response, namely the HPA axis [64,65,66,72]. Although behavioural changes could be secondary to changes in reactivity of the HPA axis, many behavioural consequences of early life programming appear to be independent of the HPA axis [71,91]. The stress-responsive dopaminergic circuitry is, therefore, also an important target, and here we shall specifically address the sexually dimorphic nature of the impact of antenatal treatment with the synthetic GC, dexamethasone.

### 3.1. Antenatal GC Treatment (AGT)

The majority of pre-clinical investigations into the effects of AGT on the brain have administered cortisol/corticosterone or synthetic GCs, such as dexamethasone, parenterally to the mother [92,93,94,95]. However, handling the animals and injection stress may, in itself, produce lasting effects on behaviour and the mesencephalic dopaminergic systems in particular [96]. An effective way to combat this is to administer GCs in the maternal drinking water [97,98,99,100]. Via this route, they are absorbed effectively, and synthetic GCs readily cross the placenta, evading the barrier of placental 11-β hydroxysteroid dehydrogenase, which inactivates cortisol/corticosterone and provides a degree of protection for the fetus. In later life, the multidrug resistance protein in the blood–brain barrier regulates the access of GCs to the brain and actively excludes dexamethasone. However, as this process is not fully developed before birth [101], GCs in the fetal circulation will also reach the fetal brain [91,102], where GC receptors (GRs), but not the mineralocorticoid receptors (also GC targets), are expressed in the basal ganglia from embryonic day 15.5 in the rat [103]. The physiological relevance of the dose of CGs administered is also a key question. In the rat, it has been estimated that the levels of endogenous GCs required for normal lung maturation during late gestation are in the range of normal, physiological stress levels and, from a pharmacokinetic standpoint, the optimal dose of dexamethasone for mimicking the late gestational elevations in GCs has been estimated to be 288 µg/kg/day over gestational days 18–20 [104,105]. Moreover, this is in the dosage range given to mothers at risk of premature delivery (circa 0.2 mg/kg/day) in order to mature fetal lungs, with great effect on infant survival [106,107,108]. Therefore, this section of the review will focus on work using a minimally intrusive treatment protocol along with a physiologically and clinically relevant dose of dexamethasone, namely, 0.5 µg, or occasionally 1.0 µg, of dexamethasone per ml of dams’ drinking water (delivering a daily dose of 150–175 µg/kg/day or 300–350 µg/kg/day) from gestational days 16-19 as a valid approach to investigating the potential translational relevance of neurobiological programming by GCs [91,98,109].

### 3.2. Macrostructure

Notable changes in regional brain volumes (either increase or decrease) in the hippocampus, amygdala, PFC and hypothalamus, indicative of structural changes, have been identified as a key feature of brain disorders [1,4,5,9]. Important correlations have been made in clinical and pre-clinical studies to suggest that, rather than being a consequence of the disorder, altered brain structural volumes may represent a pre-existing condition, likely engendered by early life stressors, which presage an increased risk for malfunction and vulnerability to psychopathology [3,9,110,111,112,113]. Antenatal treatment of non-human primates with dexamethasone, as well as a prospective study of maternal cortisol levels through human pregnancy, on changes in hippocampal and amygdala volumes, respectively, support a key role for GCs in this structural programming, which, along with associated affective problems, appear to be sex-specific [77,114,115]. There is, however, a paucity of information on volume/structural changes in the VTA/SNc regions, which are relatively small, thereby presenting a particular challenge for the resolution of in vivo volumetric data [116,117]. High resolution of small brain regions can, however, be achieved using classical ex vivo methods to assess neuroanatomical changes [91,98,109]. Using these methods, in conjunction with the AGT regimen described in Section 3.1, we have characterised sex and treatment effects on the macrostructure of the SNc and VTA in the adult rats and mice.

Dividing the nuclei into four (SNc) or three (VTA) levels relative to bregma allows for a rostro-caudal neuroanatomical segmentation of the nuclei in rats [91,118,119] and mice (Figure 1) [109], which enables an estimate of the shape as well the overall volume of each nucleus [91,109,118,120]. In control rats, the overall volumes were significantly greater in females than males by 46% for the SNc and 32% for the VTA (Figure 2). Analysis of the volumes at individual levels (A–D) also revealed a clear sex difference in the overall shape of the nuclei in the rat (Figure 2). These findings therefore add the SNc and VTA to the growing list of brain regions where innate sex differences in their total volume have been documented [121]. After AGT the shape of the nuclei were changed quite dramatically, with a tendency for an increased volume caudally (males levels C, D; females level D), but a reduced volume rostrally (females levels A and/or C). Overall, this resulted in an increase in total volume in the male SNc and VTA, as well as the female VTA, but a small, significant reduction in the female SNc volume. The net effect was a loss of the sex differences in nuclear shape and total volume, suggesting that late gestational exposure to elevated GC levels feminises/demasculinises the structure of these dopaminergic nuclei. Broadly similar effects were seen in mice, with females being generally more sensitive to a lower dose of AGT (0.5 µg dexamethasone/mL maternal drinking water) and males to the higher dose (1.0 µg dexamethasone/mL maternal drinking water) [109]. Using peripheral quantitative computed tomography scanning to assess larger brain regions, we have shown that AGT had no effect on total brain volume or relative volumes of most other brain regions, including the striatum and amygdala, except the dorsal hippocampus, where volume was decreased [122]. These results support the view that effects reported here for the SNc and VTA are true changes in regional volumes. Collectively, these findings support and expand the evidence that AGT has powerful, sexually dimorphic programming effects on brain structure. Although the concept of neuroanatomical models of disease has not previously been extended to the SNc/VTA, given current advancements in real-time in vivo imaging [123], these findings highlight the potential for SNc/VTA volume changes to be used as a sensitive, non-invasive biomarker for disorders where early life challenges, as well as dopaminergic malfunction, contribute to their aetiologies.

### 3.3. Microstructure

Macrostructural change indicates change in the cellular composition, and this will be discussed below for the dopaminergic neuronal and astrocytic populations of the SNc and VTA.

#### 3.3.1. Neurons

The SNc and VTA, classically distinguished during development as the A9 and A10 groups, respectively, of dopaminergic neurones, represent a convenient neuroanatomical separation of these neuronal populations. However, each region comprises functionally distinct sub-populations with unique projections to cortical and sub-cortical structures, albeit with a complex, non-homogeneous cellular organisation [15,24,124,125]. Thus, although no simple functional map of the midbrain dopaminergic systems has emerged, it is abundantly clear that correct organisation or positioning of dopaminergic neurons within the SNc and VTA, as well as their total numbers, critically affects their function [26,124,126]. Therefore, in order to investigate the enduring effects of AGT on cytoarchitectural organisation within the midbrain nuclei, we have adopted the rostro-caudal neuroanatomical segmentation described in Section 3.2. Using this approach, it is possible to characterise the cytoarchitectural arrangement within the SNc and VTA in terms of cellular numbers, distribution and density, using tyrosine hydroxylase immunoreactivity (TH-IR) as a marker of dopaminergic neurones, as well as autoradiographic identifications of markers of dopaminergic transmission, namely DA receptors (D1, D2) and the DA transporter (DAT).

The small number of studies where sex has been taken into consideration suggest that there are approximately 15% more dopaminergic neurones in the control male rat SNc compared with the female [91,118,127], whereas, in the VTA, females possess a greater number of dopaminergic neurones compared with males [91]. Importantly, the total numbers of dopaminergic neurones are not the only consideration, because analyses across the rostro-caudal segmentation of the SNc and VTA of control rats show that the percentage of dopaminergic populations are differentially distributed throughout the nuclei in males and females, with the distribution in males being concentrated more rostrally (level A), but more caudally in females (levels C and D) (Figure 3). Whether the innate sex differences in TH-IR cell numbers in the murine nuclei similarly reaches statistical significance remains debatable [91,127], but the distribution of dopaminergic neurones across the SNc and VTA of the mouse recapitulates the innate sex differences in rostro-caudal patterning seen in rats [109].

At the dose of 0.5 µg/mL of dexamethasone in the dams’ drinking water, AGT significantly increased the total numbers of TH-IR cells in the adult SNc and VTA of male and female rats [91,100,119]. This expanded population appeared to have the characteristics of normal dopaminergic neurones [119], as judged by their morphology, basal electrical activity and expression of Pitx3, an exclusive marker of SN and VTA neurones required for phenotype survival and maintenance [124]. Concomitant with increased cell numbers, TH-IR fibre density was also increased in the striatum, both in the dorsal regions (caudate putamen) receiving their principal input from the SNc and the ventral regions (NAc core and shell) receiving their principal input from the VTA. Although cell size was unaffected, AGT also caused a marked redistribution of dopaminergic neurones, as characterised by a reduction in the percentage located rostrally (level A) and an increase in the proportion located caudally (level C, D) (Figure 4) [91,119], clearly reflecting the rostro-caudal pattern of macrostructural shift (Section 3.2). In mice, AGT also increased TH-IR cell numbers and altered their distribution in a rostro-caudal pattern, but, in accord with the dose-dependent effects on macrostructural changes in mice (Section 3.2), the proportions of TH-IR neurons at each level were affected by the lower dose in females, but the higher dose in males [109]. These studies demonstrate that there are quantitative similarities in the effects of AGT in males and females to increase TH-IR cell numbers and also qualitative effects in that there is a rostro-caudal shift in their distribution. However, it must be remembered that these changes are imposed on a sexually dimorphic substrate (cell numbers and distribution in control animals), leading to a sex-specific cytoarchitectural re-organisation.

Investigations into factors controlling DA neurone number have largely focused on early embryonic stages. For example, it has been established that relatively early in gestation in the rat, around gestational day 10.5, the production of factors such as Sonic hedgehog (SSH) and FGF8 critically determine the induction of midbrain dopaminergic neuronal progenitors, which is fundamental in establishing the correct numbers of dopaminergic neurons in the correct position [126,128]. It would also appear that neurogenesis, specification and migration of the midbrain dopaminergic neurons are largely complete before the commencement of AGT [129,130,131]. Therefore, late gestational exposure to GCs must be influencing already committed dopaminergic neurons to alter cell numbers and intervene with the choreography of their stereotypic arrangement in the SNc and VTA at a relatively later stage of gestation (embryonic days 16–18), as discussed here, and then into early postnatal stage [99]. Interestingly, the process of up-regulation of TH expression is thought to account for a sharp rise in dopaminergic cell numbers over the last four days of gestation in the mouse [130] and their continuing postnatal rise in the SNc to adult levels at postnatal days seven–nine [132], 14 [133] or 28 [130]. As the *TH* gene contains a GC response element [134] and GRs are expressed in the rat basal ganglia by embryonic day 15.5 [103], it is possible that AGT could influence terminal differentiation to the DA phenotype. Naturally occurring cell death via apoptotic mechanisms is also a process thought to play a critical role during late gestation and the neonatal period in regulating adult numbers of dopaminergic neurons in the SN [133,135,136,137]. Figure 4 shows that GC exposure on gestational days 16–19 markedly reduced apoptotic markers selectively in the TH-IR cells of the SNc and VTA by post-natal day two [109]. These findings accord with the ability of GCs to promote the survival of hippocampal granule cells in the developing dentate gyrus [138,139] and suggest that AGT-mediated suppression of the wave of ‘classical’ neuronal programmed cell death may be a mechanism contributing to the enduring change in dopaminergic cell numbers in the adult SNc and VTA.

In terms of neurochemical parameters, autoradiographic binding densities for D1, D2 and DAT were similar in control male and female rats in the sub-cortical and cortical regions, except for D1 in the CPu, which in males was double that seen in females. Baseline extracellular levels of DA were also similar in control males and females. However, in control males DA efflux in response to amphetamine, which binds to DAT and is commonly taken as an indicator of dopaminergic tone [140], was only 25% of that seen in females (Figure 5), suggesting marked sex differences in DA terminal dynamics. This may accord with the typically greater locomotor effects of amphetamine in females, and adds to other evidence of inherent sex dimorphisms in the ascending dopaminergic systems [73,74]. It is also plausible that the raised levels of D1 receptor binding in the control male CPu could counter any apparent reductions in DA neuronal dynamics and theoretically preserve basal ganglia output, which relies heavily on D1-mediated excitability of striatal medium spiny neurones [141].

AGT induced marked region- and sex-specific changes in the neurochemical markers of DA transmission (Table 1) [119]. Consistent between the sexes was an increase in binding densities for D2 in all three sub-cortical regions (caudate puramen (CPu), NAc core and shell), but no effects were seen in the infra-limbic cortex (ILC) and the pre-limbic cortex (PLC). D2 receptors may be located pre-synaptically, where they exert autoinhibitory control on DA release, or post-synaptically. While autoradiographic data cannot distinguish between the two locations, the PFC (prefrontal cortex) is thought to be devoid of pre-synaptic D2 receptors [142], supporting the idea that the striatal AGT-responsive receptors are likely to be pre-synaptic. An increase in D2 auto-inhibitory control in the striatum may, therefore, be a compensatory response which could account for the fact that striatal extracellular levels of DA remain unaffected by AGT [119], despite the substantial increase in dopaminergic innervation, which would be expected to lead to a hyper-dopaminergic state.

Unlike effects on D2 receptors, AGT had diametrically opposite effects on D1 receptors in male and female rats, substantially reducing binding densities in males (CPu, NAc shell and PLC), but increasing them in females (CPu, NAc core, ILC and PLC). Interestingly, amphetamine-induced DA efflux was enhanced four-fold in AGT-males, but was reduced in AGT-females, raising the possibility that the respective, co-incident up-regulation and down-regulation of post-synaptic D1 receptors may represent compensatory changes to preserve DA transmission. In parallel with this, AGT increased DAT level in males (CPu, NAc core), but decreased them in females (CPu, NAc core and shell, ILC, PLC), which would also be in accord with sex differences in the direction of AGT-induced changes in sensitivity to amphetamine (Figure 5). As a pre-synaptic marker, the DAT protein has also has been used as an indicator of terminal density. Accordingly, an increase in striatal DAT binding in AGT-males is in line with the AGT-induced TH-IR terminal density. However, this argument does not hold true in females, where DAT density is reduced after AGT, despite an increase of ~40% in TH-IR terminal density. This apparent discrepancy may simply reflect the importance of DAT in adapting to prevailing conditions, and cautions against its sole use as an indicator of terminal density. Additionally, counter-regulatory influences from the PFC should also be considered.

The mesocortical DA system interacts with pyramidal and non-pyramidal cortical neurons expressing D1 and D2 receptors [143] and exerts powerful inhibitory control over the mesolimbic DA system [144,145]. As summarised in Table 1, in male progeny, AGT had no influence on synaptic markers of DA transmission (D1, D2, DAT) in the IL cortex [9], which projects to the NAc shell [146,147], whereas D1 levels in the PL cortex (projecting to NAc core) were strongly reduced. This suggests a loss of mesocortical inhibitory control of the mesolimbic system after AGT, which accords with the enhanced mesolimbic activity revealed by amphetamine-induced DA efflux in AGT-exposed males. Conversely, in AGT-exposed female progeny, a dramatic increase in PL and IL D1 levels and a precipitous fall in DAT to almost negligible levels [119] could indicate an increased inhibitory drive from the PFC, compatible with reduced mesolimbic DA activity revealed by amphetamine administration. These data demonstrate that mesocortical as well as mesolimbic DA networks are differentially influenced by AGT in males and females. As corticostriatal axons can facilitate DA release at terminal level without affecting DA impulse activity in the VTA [148], altered mesocortical DA activity could explain why AGT altered intrinsic mesolimbic DA activity without affecting the neurophysiology of VTA neurons in adult rats [119].

#### 3.3.2. Astrocytes

Astrocytes are the main glial cell type in the brain and, far from the original view that they are the inert packing material providing physical support for neurons (the cell types often regarded as the principal functional entities in the brain), we now know that astrocytes play critical roles in normal brain development, structure and function as well as pathologies [149,150,151]. However, the contribution of this cell type to neurobiological programming in the brain has been largely ignored, but studies in mice [109] and rats [152] have demonstrated profound effects of AGT on the number, density and distribution of astrocytes in the SNc and VTA, depending on dose, sex and sub-region of the SNc/VTA. In control animals, the total number of cells expressing immunoreactivity for glutamine synthetase (GS-IR), the astrocytic marker ([150,153], and their distribution across the nuclei were similar in males and females, except for the near negligible numbers at the midpoint of the female VTA (level C, Figure 6). At a dexamethasone dose of 0.5 µg/mL in the dams’ drinking water, AGT produced a dramatic two–three-fold increase in the estimated total numbers of GS-IR cells of the adult SNc and VTA of both sexes (greater in females than males), with effects at most levels, depending on sex. Perhaps surprisingly, this was not accompanied by any volume changes, thereby leading to a marked two–four-fold increase in cell density at all levels throughout both nuclei, except for the female VTA at level C, where density increased seven-fold. This reveals that cellular changes can occur without structural changes, highlighting that structural changes alone may not be sufficient as potential indicators of deficits in function, which would also require markers of functional and/or biochemical change. Using a higher dose of AGT (1.0 µg/mL), GS-IR cells appeared largely unaffected in AGT-females. A broader dose–response curve would be needed to investigate further, but this may well reflect a bell-shaped curve typically seen for the actions of GCs. In contrast, in males, astrocyte counts were similarly raised after both the high and low doses, but unlike the low dose, the higher dose led to an expansion of regional volumes, resulting in no effect on astrocyte density. Further studies measuring astrocyte size would be invaluable in interpreting these findings, but this is a complex matter. It is not clear what factors regulate astrocyte size, but this is crucial to their function as each astrocyte occupies a large, unique spatial domain associating with several hundred dendrites and 100,000 synapses in the rodent brain, and 20-fold more synapses in the human brain [154]. Although GS-IR is very effective for visualising the immediate astrocyte cell body, making it ideal for cell counting purposes, it provides little information on astrocyte size, which is a very dynamic entity. Another commonly used astrocytic marker, glial fibrillary acidic protein (GFAP), detects an astrocyte-specific cytoskeletal protein, but this marker also has its limitations in that it has been estimated that it reveals approximately only 15% of total astrocytic volume [150,154]. Whatever the precise effects of AGT are on astrocyte structure and their overall contribution to the altered volume and shape of the adult SNc and VTA, it is clear that AGT alters the astrocytic environment in which the dopaminergic neurons function in a sexually dimorphic manner. This suggests a further mechanism which may underpin the sex-specific effects of early environmental disturbances on adult behaviours, and supports the view that astrocytic disruptions contribute to DA-dependent CNS pathologies which exhibit sex differences as well as a developmental component, including schizophrenia, depression and neurodegenerative disorders [149,150,151].

The effect of AGT at a dexamethasone dose of 0.5 µg/mL in the dams’ drinking water to increase astrocyte cell density was manifest by P2 in both sexes [119]. As most astrocytes are produced post-natally [155] and cessation of AGT at day 19 of a 21 day gestation period ensures that the mice are dexamethasone-free at birth, it follows that postnatal astrocyte development is highly sensitive to prenatal events, the course of which can be altered by AGT. Neural precursor cells in the developing vertebrate CNS first form neurons and then undergo a “neurogenic-to-gliogenic switch” late in gestation [154], but the triggers for this switch are unclear. Notably, GRs are expressed in astrocytes, GCs can up-regulate GS, and maturation of the adrenal gland, with the associated rise in GC levels, has been linked to glial differentiation, at least in the chick retina [153,156]. The sensitivity of the SNc/VTA astrocytes to AGT further indicates an important role for GCs in astrocyte ontogeny and invites the intriguing speculation that GCs have a role in regulating neural precursor cell fate.

#### 3.3.3. Behaviour

As reviewed elsewhere [58], there are many inconsistencies in the findings from behavioural studies regarding the responses of the mesocorticolimbic and nigrostriatal dopaminergic systems to intrauterine stressors, with outcomes that may be detrimental, advantageous or without effect, and it is often assumed, but not proven, that exposure to maternal GCs effects the change [157]. The reasons for these inconsistencies will be due inevitably to many variations in experimental design [9], which are likely to present different challenges and elicit unique responses from the dopaminergic systems. The considerable variation in the nature, intensity and duration of the chosen stressors is one key factor and, although cortisol release will be a common response, each stressor may also challenge other systems in the body, affecting, for example, immune, metabolic and cardiovascular responses, adding their unique influences to those of the HPA axis. Outcomes will also depend on the degree of maturity of the neural substrate at the point of exposure to the challenge as well as the age and sex of the progeny when behavioural tests are performed [9,68,70,71,83,158,159]. Here, we shall focus on the consequences of AGT given during late gestation on performance in behavioural tests known to depend on midbrain dopaminergic circuitry, which has relevance for GC use in perinatal medicine, as well as some bearing on late gestational stressors.

Taken individually, one could reasonably predict behavioural disruption by AGT on the basis of the substantial neurobiological changes that are seen in DA cell counts, striatal fibre density, DA receptor and DAT levels, and DA release (Section 3.2, Section 3.3.1 and Section 3.3.2). Accordingly, when rats were placed in a novel environment, spontaneous open-field locomotor activity was attenuated in female progeny after AGT (Figure 7A,B), indicating suppression of motivational arousal, which depends on DA release within the mesolimbic and nigrostriatal DA systems [160]. In contrast, males were unaffected in this test (Figure 7A,B). Interestingly, these effects are compatible with the co-incident, sexually dimorphic neurobiological changes, which indicated a reduction in intrinsic mesolimbic dopaminergic activity in AGT-females, whereas in AGT-males the increase in activity may be offset by an increase in DAT and a reduction in D1 receptors in the striatum [119]. Using a completely different challenge of intrauterine anoxia stress, male rats showed an increase in spontaneous locomotor activity (a hyper-activity of the mesolimbic-NAc system), which was attributable to a loss of inhibitory control from the medial PFC, possibly due to an increase in DAT, but with no effect on D1 and D2 receptors in the PFC [40]. These contrasting outcomes stress the susceptibility of the dopaminergic systems to neurobiological programming and also serve as a clear illustration of the great variability in outcomes, depending on the nature of the challenge. In male rats, a startle stimulus has been shown to lead to a fall in NAc DA levels, but if a weak pre-pulse is presented prior to the startle stimulus, the fall in NAc DA is attenuated [161]. In contrast to the effects of AGT on spontaneous open-field activity, the performance of AGT-females was unaffected in this test of pre-pulse inhibition (PPI; Figure 7D), whereas in males PPI was exaggerated (Figure 7C), which would be compatible with greater response to amphetamine in AGT-males (Section 3.3.1, Figure 5) [119]. Surprisingly, however, in other established tests of mesolimbic activity, including amphetamine-induced locomotor activity, cocaine self-administration and learning in response to appetitive cues predictive of food [160,162,163,164,165], neither males nor females were affected by AGT [119]. We conclude, therefore, that apparent behavioural normality in some, although not all, circumstances is achieved by powerful, AGT-induced compensatory mechanisms, which preserve DA transmission via different, even opponent, processes in males and females (as indicated by the sexually dimorphic neurobiological changes consequent on AGT). Our recent study shows that compensatory mechanisms also operate within the mesocortical dopaminergic system. Thus, we found that AGT dramatically protects male rats from the disruptive effects of a D1 receptor agonist on a test of spatial working memory, a cognitive function that depends on the functional integrity of DA inputs to the mPFC [166].

It must be noted, however, that some studies involving late gestational GC treatment, have reported significant behavioural effects in both sexes, especially in tests of anxiety and depression-like behaviours [70,93,94,95]. However, most studies include the additional stress of parenteral administration of GCs during gestation, and often high doses of dexamethasone, several-fold greater than that used in perinatal medicine, are administered [93,95]. Additionally, the tests of anxiety and depressive-like behaviours, such as the forced swim tests or the elevated plus maze, encompass stress exposure in adulthood [93,95]. This contrasts with the behavioural tests described in Virdee et al. [119], where the aim was to investigate animals under basal conditions, reflecting intrinsic functional connectivity [167]. Such differences are likely to have important consequences for behaviour, because apparently protective adaptations that allow behaviours to appear normal under basal conditions, may fail when confronted with challenges later in life. In support of this, human and non-human primate studies reveal abnormalities in the HPA response to stressors in adult life in individuals exposed to early adversity [168,169] and the central dopaminergic response to stress in adult guinea-pigs and rats is altered in animals exposed to birth insults [170]. Furthermore, mesolimbic dopamine release in response to a psychosocial stressor in human adults is sensitive to the effects of early life events [75]. Collectively, these observations suggest that, although the adaptive operational changes in the midbrain dopaminergic systems are protective within certain limits, this is at the cost of the network operating outside its normal limits, a state termed allostasis, or ‘stability through change’ [171,172], so that an apparent resilience to early adversity could, ultimately, become a pre-disposing factor to (psycho)pathology.

### 3.4. Why Should the Influences of AGT Be Sexually Dimorphic?

Some subtle differences in the rates of maturation of male and female foetuses may have implications for sex-specific programming influences [76], but most research attention has focused on genetic and hormonal influences. By late gestation, male and female embryos have experienced a substantially different genetic and hormonal milieu, which is critically responsible for survival of the species. This underpins sex differentiation not only of the male and female reproductive tracts, but also of hypothalamic circuitry controlling ovulation in females as well as unique reproductive behaviours that characterise females and males [73,74,85,173,174,175,176]. At conception, the embryo is sex neutral, but after a few days’ gestation in rats and mice, or around six weeks’ gestation in humans, expression of genes on the male Y chromosome initiate processes of masculinisation and defeminisation. Specifically, activation of the sex-determining region of the Y chromosome (*SRY*) gene directs development of male genitalia, and activation of the anti-mullerian hormone (*AMH*) gene blocks the formation of female genitalia. The associated early gestational wave of testicular testosterone production stimulates maturation of the male gonads. A later wave of testosterone production occurs in the rat at embryonic days 17–19 and immediately after birth [177,178] or around mid-gestation in humans [179]. This is a critical window for brain development, when masculinisation/defeminisation of the male hypothalamus occurs, without which this circuitry would proceed largely along female lines by default. However, testosterone readily crosses the blood–brain barrier to reach all brain regions, where it has the potential to permanently modulate neurite extension, synaptogenesis, cell survival and apoptosis. Hence, growing evidence suggests that many other brain regions, such as the hippocampus, amygdala and cortex, which are involved in memory, learning, cognition, emotion and stress reactivity, are subject to such hormonal sex differentiation at critical windows of development [9,85,175,180,181,182,183]. Raised GC levels and intrauterine stressors are, therefore, acting on a significantly different substrate in male and female brains, providing a framework to explain the sexually dimorphic consequences of AGT. In support of this, perinatal stressors and elevated GCs can interfere with normal masculinisation of the hypothalamus and male sexual behaviour [177,184,185,186], and our findings that AGT can feminise the structure, shape and cytoarchitectural arrangement within the SNc and VTA, would support and extend this concept to other brain regions.

However, direct evidence is required to confirm that the innate sex differences seen in the structure and function of the SNc and VTA (Section 3.2, Section 3.3.1 and Section 3.3.2 [73,74]) are indeed, hormonally programmed during the classical period for brain sexual differentiation. Using the classical hormonal manipulations of gonadectomy of newborn male rats and testosterone treatment of newborn females in order to achieve, respectively, the removal or addition of the masculinising/defeminising influences of testosterone [187], we have attempted to provide this evidence for the organisational (irreversible) effects of sex hormones on the SNc and VTA systems. In view of the sensitivity of these systems to neurobiological programming, it is, perhaps, not surprising that we found that the sham-operated and vehicle-injected control animals showed marked changes in TH-IR cell numbers in the VTA and SNc, making the data uninterpretable (unpublished observations). In the absence of such confirmatory data, one should consider a role for the activational (reversible) influences of the adult sex hormone environment. Studies in gonadectomised rats have shown that estradiol in females enhances dopaminergic transmission, whereas neither estradiol nor testosterone has any influence in males [188,189,190,191]. Therefore, as responses are clearly still sexually-differentiated when the influences of sex hormones are removed and equalised in males and females, it follows that there remains a significant sex difference in the underlying circuitry. Moreover, sex hormone manipulations in adulthood do not affect TH-IR cell number, the prevailing differences in the adult hormonal environment cannot account for sex differences in cell numbers [118,192,193]. Together with the sexually dimorphic responses of the midbrain systems to environmental disturbances in utero already discussed, these findings constitute strong indirect evidence in support of a role for hormonal-mediated organisation of these sex differences.

Although sex hormones are considered to be the primary factors for driving structural and functional sex differentiation of the brain, there is evidence to suggest that genetic, cell-autonomous factors also make a contribution. For example, investigations using sex-specific mesencephalic cultures derived from embryonic day 13, a time-point presumed to precede the late gestational rise in testosterone, reported the emergence of sex-specific characteristics in terms of TH-IR cell numbers, DA levels and DAT activity in rats and some, but not all, strains of mice, suggesting an influence of genetic background [194]. Evidence in humans and other species suggest that the sex chromosomes themselves are likely contributors to such sex differences [195], especially the *SRY* gene. Although the actions of this gene were classically thought to be restricted to sex determination early in gestation, its expression has now been identified through development and into adulthood in a number of male non-reproductive tissues, including the brain in humans, rats and mice [127,196,197]. Moreover, SRY expression co-localises with a subset of dopaminergic neurones of the SNc in adult rodent and human post-mortem brains [127,198] and may contribute not only to the sex differences seen in dopaminergic cell numbers, but also the greater male susceptibility to PD [199]. Whether a similar impact of *SRY* occurs in the VTA dopaminergic populations remains to be discovered. Clearly, future studies are needed to determine how sex-specific genes, as well as hormonal factors, differentially alter interactions of DA populations with the early environment and susceptibility to later disease in males and females.

## 4. Summary

Neuroanatomical and neurochemical parameters clearly demonstrate that intrauterine exposure to inappropriately raised levels of GCs has profound effects on the normal developmental trajectories of the midbrain dopaminergic systems, leading to enduring changes in the cytoarchitecture (neurons and glia) and activity of the mesocorticolimbic and nigrostriatal dopaminergic systems. Moreover, the changes in these parameters are often qualitatively diametrically opposite in males and females. Remarkably, these robust, sexually dimorphic changes were not accompanied in either sex by altered psychomotor and appetitive behaviours that are known to depend on midbrain dopaminergic transmission. We propose, therefore, that such neurobiological changes represent a broad spectrum of enduring adaptive mechanisms, which underpin apparent behavioural resilience to early environmental perturbations, but these proceed via very different mechanisms in males and females. However, such adaptations appear not to be completely protective. For example, certain DA-dependent behaviours can be affected by AGT in females, but not males (locomotor response to novelty), whereas others are affected in males, but not females (sensorimotor gating). Hence, AGT-induced permanent adaptions in dopaminergic circuitry proceed via different, often opponent, mechanisms in males and females. This establishes an important principle of neurobiological programming that is likely to hold for all types of early-life challenges affecting all brain systems, whether dopaminergic or not [82,200]. Although a rather neglected area, recent evidence also suggests a key role for glial cells in the sex-specific process of biological programming, which merits closer investigation.

Whatever the underlying processes, AGT programming of the midbrain dopaminergic systems has significance for the use of GCs in perinatal medicine. In around 10% of pregnancies worldwide, women are given AGT to promote foetal lung maturation and infant survival in cases of threatened premature delivery. A single course of AGT has unquestionable value, with little evidence for deleterious effects [107]. However, repeated courses, which have been given routinely to women who remain undelivered (50% of cases), are linked to long-term, potentially deleterious effects on the developing brain [201,202,203]. These include decreased bodyweight, head circumference and brain volume at birth [204], altered stress responses in infants [66,205] and altered behaviours, such as hyperactivity and distractibility [10], which have been linked to alterations in dopaminergic signalling [12]. Due to the complex and varied nature of stressors, it would be unwise to suggest that the outcomes discussed here for AGT-induced programming of the brain would be identical to those incurred after intrauterine exposure to stress. However, human studies have made direct links between stress-induced elevations in maternal cortisol levels outside the critical, late-gestational window when levels normally rise, and sexually dimorphic effects in the brain. These include sex-specific changes in regional brain volumes and HPA axis reactivity, as well as compromised aspects of cognition and affective behaviour in childhood [63,115,206,207]. Systematic investigations into the potential impact of GCs on the developing brain and neurological and behavioural outcomes are, therefore, key to ensuring their safe use clinically.

In summary, the evidence presented here has identified sexually dimorphic capacities for molecular adaptations within the midbrain dopaminergic systems in response to AGT, which may confer sex-specific behavioural resilience or vulnerability to early-life environmental challenge, depending on the specific behaviour tested. Moreover, the sex dimorphisms may have translational relevance for DA-dependent psychopathologies that show a sex bias as well as a susceptibility to early environmental challenge [208]. For example, we have identified an effect of AGT on female, but not male, behaviours (motivational arousal), which have correlates with behaviours that are altered in depression, a condition generally more prevalent in women [209]. Equally, we have highlighted male behaviours (pre-attentional processing/PPI), which are affected in male, but not female, schizophrenics [210]. Notable sex differences in the capacity to adapt to early life challenges may, therefore, underpin the notable sex bias in DA-dependent psychopathologies, and highlights the need for a greater understanding of the underlying processes if we are to improve treatments, which need to be tailored to the specific needs of men and women. A key challenge for the future will be the elucidation of mechanisms that underpin the sexually dimorphic responses to AGT. Epigenetic programming is emerging as a critical factor in brain sexual differentiation, which is driven, to a large extent, by a surge in testosterone production by the testes, occurring between GD17 and postnatal day 10 in the male rat [211,212]. Prenatal synthetic GC treatment can also permanently modify the epigenome [213], and produce endocrine and behavioural effects which may be qualitatively and quantitatively different in males and females [12,65,81]. Differential interactions of glucocorticoids with the sex-specific gonadal steroid environment at the level of epigenetic markers in the developing male and female brain provides a platform for sexually dimorphic outcomes [214]. Whether the midbrain DA systems are targets for such actions remains to be determined, but this would offer a compelling explanation for their sexually dimorphic programming by AGT.

## Figures and Tables

**Figure 1 brainsci-07-00005-f001:**
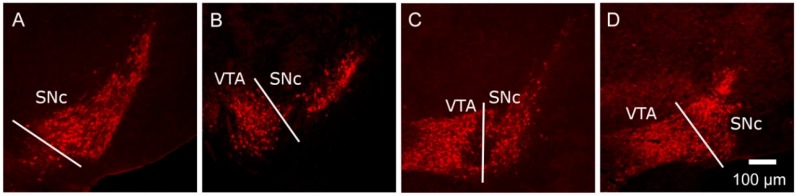
Representative images, moving in a rostro-caudal direction, at the different levels of the murine SNc (substantia nigra pars compacta) (**A**–**D**) and VTA (ventral tegmental area) (**B**–**D**). Brain slices were processed for tyrosine hydroxylase immunoreactivity (TH-IR) and the SNc and VTA were delineated by the TH-IR cell bodies and classical neuroanatomical landmarks. In order to detect any regional differences in shape/volume (calculated using Cavalieri’s method [91,109] and cell distribution through the nuclei, sections containing the nuclei were divided into four levels (**A**–**D**), each spanning 200–250 µm, beginning at the anatomical level where the SNc TH-IR cells first appear. Representative images at each level are shown here for the mouse, with full details given in [109]. This method is analogous to that which we have used previously for analysing the rat SNc/VTA [91].

**Figure 2 brainsci-07-00005-f002:**
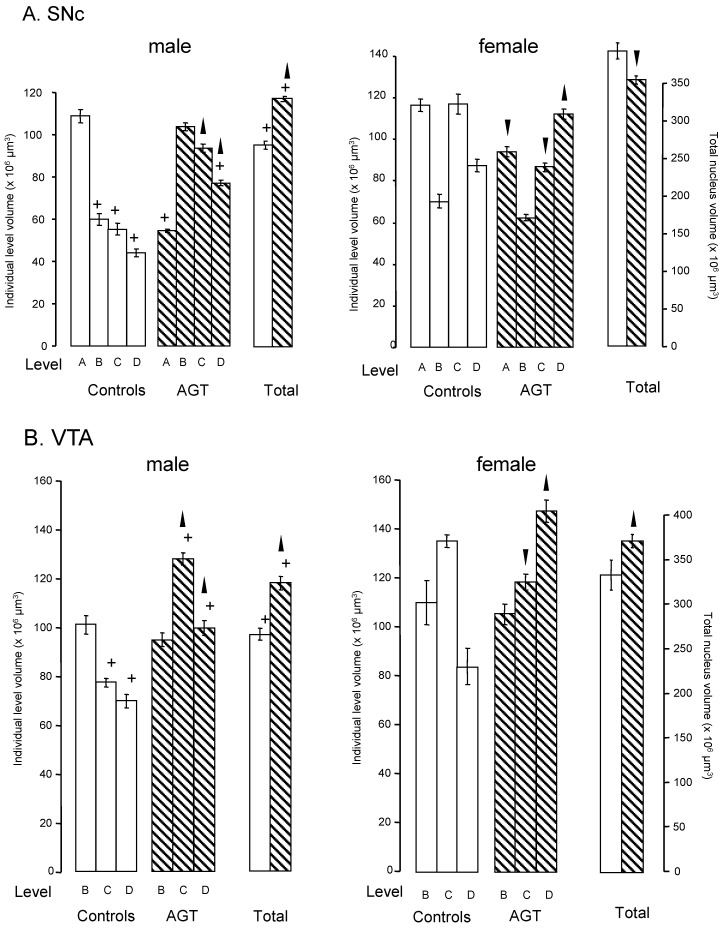
Influence of antenatal glucocorticoid treatment (AGT) on the volume (levels A–D and total) and overall shape of the adult rat SNc and VTA. Rats were exposed to AGT (dexamethasone via the maternal drinking water prenatally on embryonic (0.5 μg dexamethasone/mL in dams’ drinking water on gestational days 16–19, 
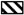
) and volumes of the SNc (**A**) and VTA (**B**) in adulthood were compared to the control offspring of dams receiving normal drinking water (
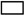
). Data are means ± s.e.m., *n* = 8 animals per treatment group. 
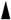
, 
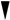
 indicates significant effect of treatment, *p* < 0.05, increased or decreased respectively, for dexamethasone treated vs. control animals; + indicates significant sex difference *p* < 0.05 vs. females in the same treatment group.

**Figure 3 brainsci-07-00005-f003:**
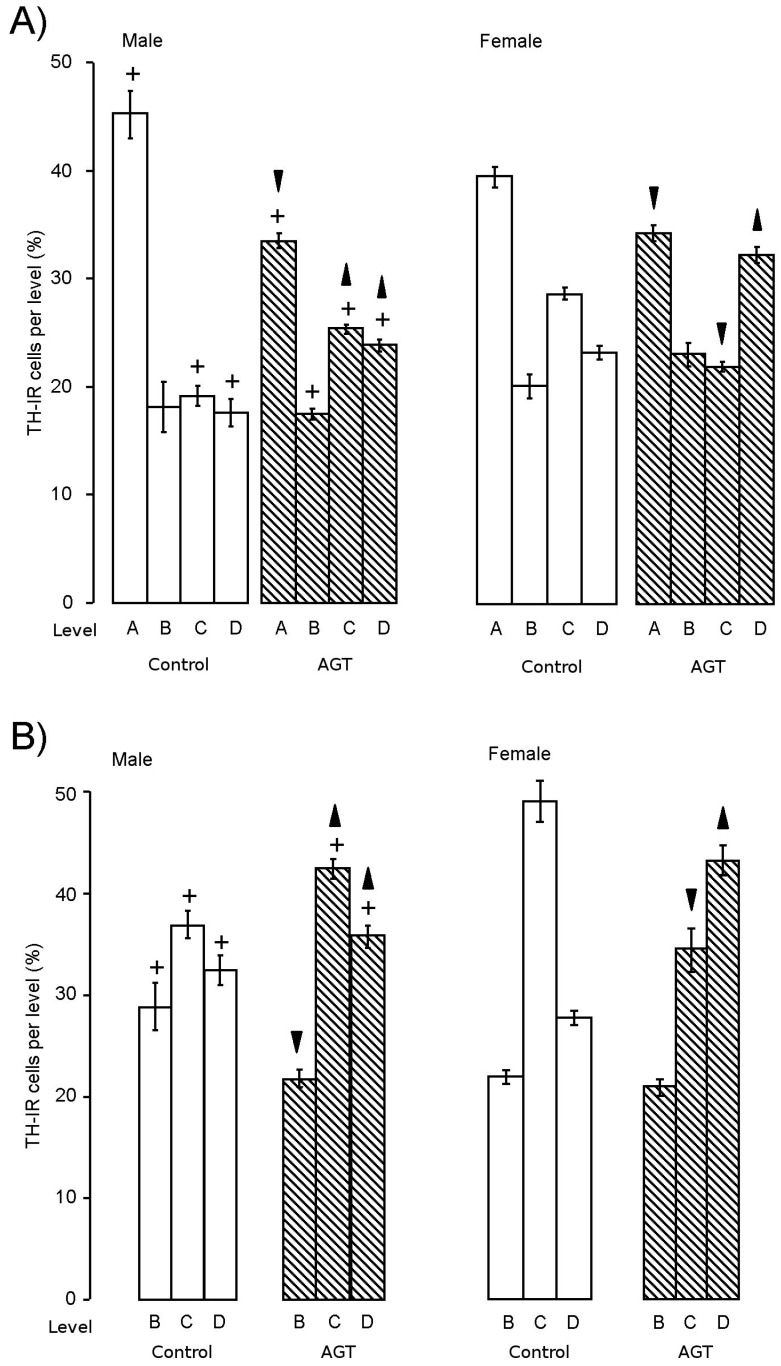
Influence of AGT on the redistribution of TH-IR (tyrosine hydroxylase immunoreactivity) cells. The percentage of the total number of TH-IR cells located at each level throughout the SNc (**A**) and VTA (**B**) was calculated for adult male and female rats after antenatal treatment with dexamethasone via the maternal drinking water on embryonic days 16–19 (0.5 μg/mL, 
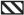
) compared to the control offspring of dams receiving normal drinking water (
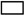
). Data are means ± s.e.m., *n* = 8 animals per treatment group. 
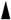
, 
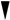
 indicates significant effect of treatment, *p* < 0.05 increased or decreased, respectively, for dexamethasone treated vs. control animals; + indicates significant sex difference *p* < 0.05 vs. females in the same treatment group.

**Figure 4 brainsci-07-00005-f004:**
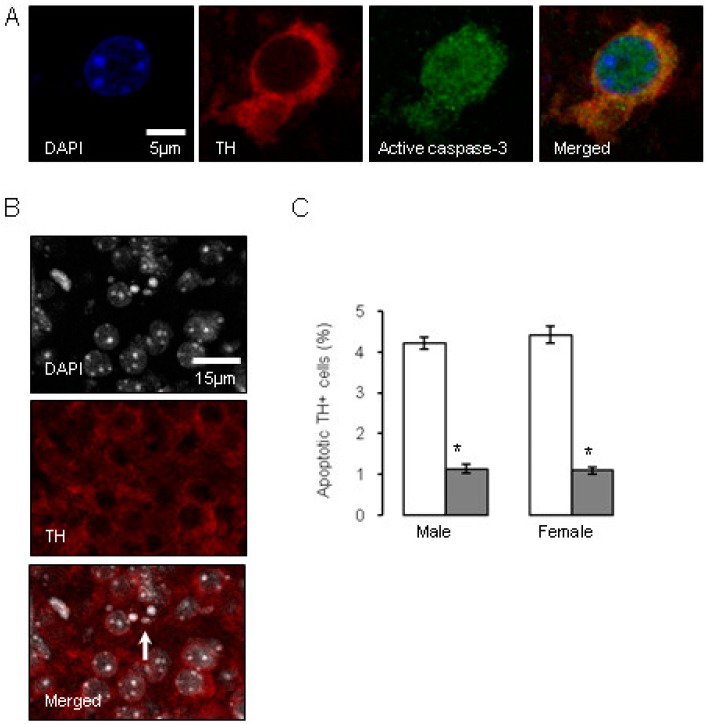
Influence of antenatal glucocorticoid treatment (AGT) on apoptosis in dopaminergic neurons (TH-IR cells) on postnatal day 2 (P2). A proportion of murine TH-IR cells undergo apoptotic changes on postnatal day P2, as identified by immunostaining for activated caspase-3 (**A**), or by the presence of nuclear condensation (**B**). The white arrow indicates an example of pronounced nuclear condensation. TH-IR cell apoptosis was quantified in the combined SNc and VTA at postnatal day P2 in male and female mice (**C**) treated with dexamethasone via the maternal drinking water antenatally on embryonic days 16–19 (
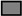
 0.5 µg/mL), and compared to the control offspring of dams receiving normal drinking water (
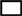
). Data are means ± SEM, *n* = 6 animals per treatment group. * Indicates a significant effect of treatment, *p* < 0.05 for dexamethasone treated vs. control animals.

**Figure 5 brainsci-07-00005-f005:**
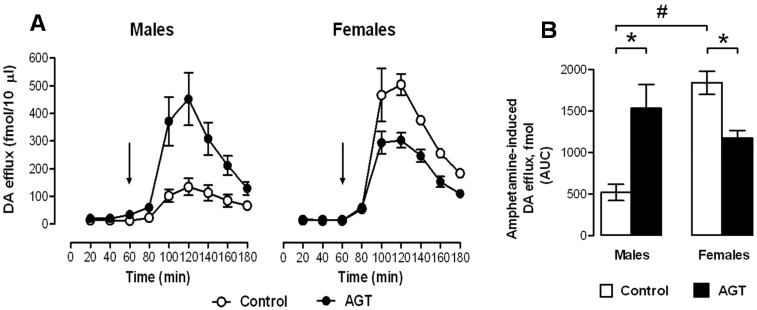
Impact of AGT on the neurochemical response to amphetamine. Male and female rats exposed to antenatal glucocorticoid treatment (AGT; dexamethasone, 0.5 µg/mL, in dams’ drinking water on gestational days 16–19) and controls (dams received normal drinking water) were tested in adulthood. Extracellular levels of DA (dopamine) in the NAc (nucleus accumbens) were assessed by in vivo microdialysis coupled with electrochemical detection; samples were collected every 20 min. (**A**) the line plots depict DA levels in each 20-min fraction for controls (pen circles) and AGT subject (solid circles); the arrow indicates the point of amphetamine administration (0.8 mg/kg *i.p*.); (**B**) The bar plot shows cumulative DA release above baseline (area under the curve, AUC). Values represent means ± s.e.m for control males (*n* = 5), AGT males (*n* = 6), control females (*n* = 4) and AGT females (*n* = 5) (*n* values vary according to correct placement of dialysis probe and statistical analyses were adjusted accordingly). # *p* < 0.05, indicating a significant sex difference; * *p* < 0.05 indicating a significant effect of AGT. For full details see [119].

**Figure 6 brainsci-07-00005-f006:**
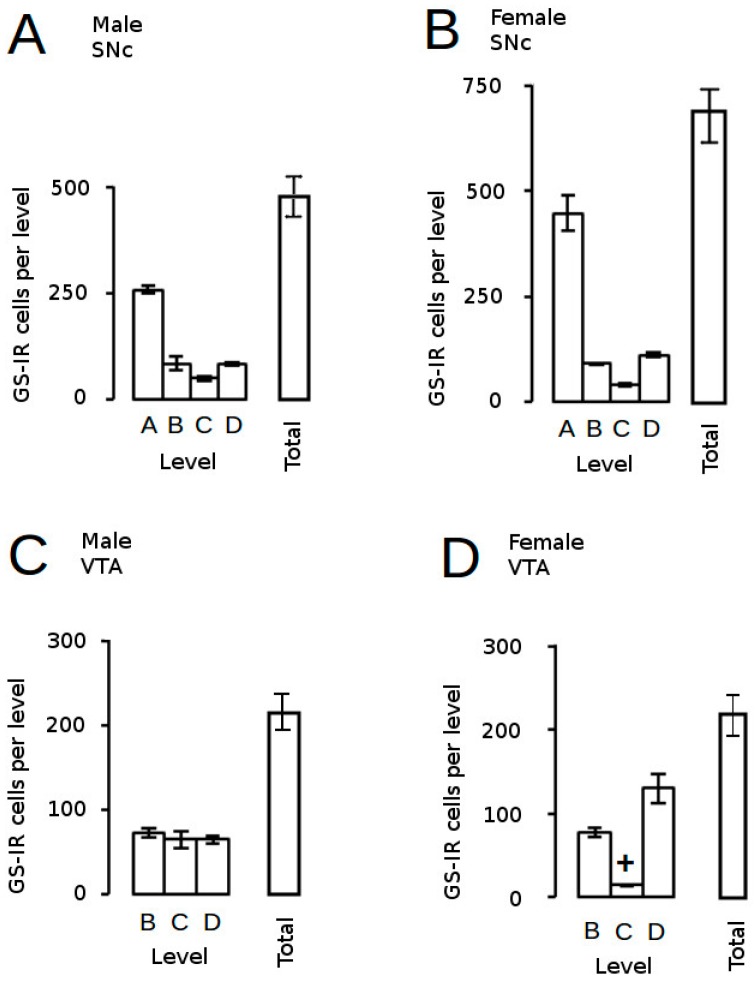
Estimated astrocyte counts in control rats. The distribution of astrocytes (GS-IR, immunoreactivity for glutamine synthetase) across the adult SNc (**A**,**B**) and VTA (**C**,**D**) and estimated total astrocyte counts are presented for male (**A**,**C**) and female (**B**,**D**) mice. + *p* < 0.05 for male vs. female. For fill details see [109].

**Figure 7 brainsci-07-00005-f007:**
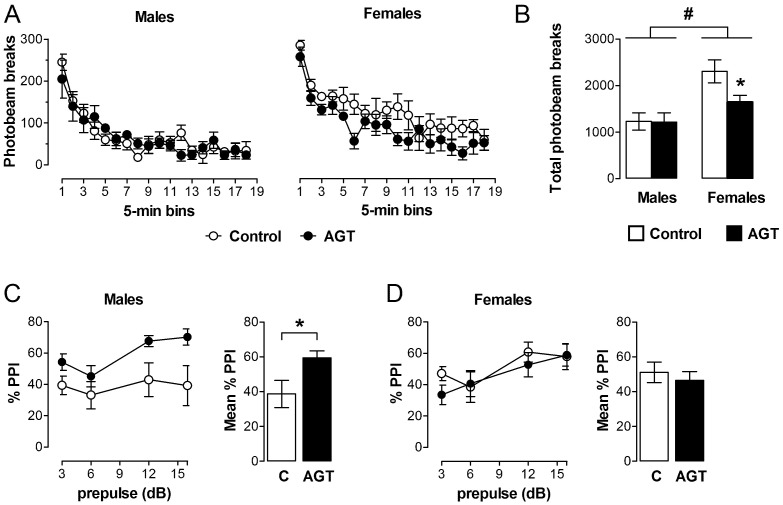
Behavioural impact of AGT. Male and female rats exposed to antenatal glucocorticoid treatment (AGT; dexamethasone, 0.5 µg/mL, in drinking water on gestational days 16–19; solid circles/bars) and controls (dams received normal drinking water; open circles/bars) were tested in adulthood. (**A**,**B**) Spontaneous locomotor activity is suppressed in adult female, but not male, progeny after AGT. Data are mean ± s.e.m., *n* = 6; * *p* < 0.05 for AGT vs. control group; # main effect of sex, *p* = 0.001. (**C**,**D**) The paradigm of prepulse inhibition of the startle response (PPI) was used to investigate sensorimotor gating in young adult offspring. This is significantly modulated after exposure to AGT in adult male (**C**), not female (**D**), progeny. Data are mean ± s.e.m., *n* = 8; * *p* < 0.05 for AGT vs. vehicle. For full details see [119].

**Table 1 brainsci-07-00005-t001:** Summary of autoradiographic analysis of changes in dopaminergic signalling proteins after antenatal glucocorticoid treatment (AGT).

Binding Densities	Controls Males vs. Females		AGT Males (m) vs. Same Sex Controls					AGT Females (f) vs. Same Sex Controls				
	CPu	NAc	CPu	Nac Core	Nac Shell	ILC	PLC	CPu	Nac Core	Nac Shell	ILC	PLC
D1	m > f	ns	**↓**	ns	**↓**	ns	**↓**	**↑**	**↑**	ns	**↑**	**↑**
D2	ns	ns	**↑**	**↑**	**↑**	ns	ns	**↑**	**↑**	**↑**	ns	ns
DAT	ns	ns	**↑**	**↑**	ns	ns	ns	**↓**	**↓**	**↓**	**↓**	**↓**

Foetal rats were exposed to AGT (dexamethasone 0.5 µg/mL in dams’ drinking water on gestational days 16–19). In adulthood (3 months of age) the offsprings’ brains were analysed for the expression of dopamine receptors (D1, D2) in the striatum (caudate putamen, CPu, and nucleus accumbens, NAc, core and shell) and the pre-frontal cortex (infralimbic cortex, ILC, and pre-limbic cortex, PLC). **↓** and **↑** represents significant decreases or increases, respectively, for the effects of AGT relative to same sex controls; ns indicated no significant effect. Male/female comparisons showed significant sex differences for D1 and DAT (dopaminergic transmission) binding across all regions after AGT. Full details and data can be found in [119].

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
