# Peer review of "Enduring, Sexually Dimorphic Impact of In Utero Exposure to Elevated Levels of Glucocorticoids on Midbrain Dopaminergic Populations"

_brainsci, 2016, doi:10.3390/brainsci7010005_

Round 1

Reviewer 1 Report

Summary. This is a review paper that considers the role of midbrain dopaminergic systems in mediating the effects of prenatal glucocorticoid exposure on fetal development. Although human evidence is cited, the primary basis for this paper is a model of antenatal glucocorticoid exposure that is designed to mimic the ‘real world’ conditions of human antenatal glucocorticoid exposure. The conclusion is that AGT-inducted permanent adaptations in dopaminergic circuitry proceed via different, often opponent mechanisms in males and females.

Review. The paper is very well-written and argued. The evidence is well-organized and sufficient to support the conclusion. I have only minor comments for the author’s consideration in preparation of a revised manuscript.

Line 77. The word ‘prove’ really jumps out because the data to date is not definitive. I think that “leads to the conclusion” would be acceptable, because the available evidence is strong, but proof implies that there is definitive evidence.

Line 80. The word ‘malfunction’ seems out of place and non-specific. Later in the paper specific example of malfunction are provided, and those seem fine, however here a different word should be chosen or further context should be provided to define what sort of malfunction is meant.

Line 213. The sentence there implies that an underdeveloped blood brain barrier is responsible for the movement of GCs into the fetal brain, however the diffusion or transport of GCs across the BBB is essential to the effects of the HPA axis on brain function and is not limited to the fetus. See for example Mason BL, Pariante CM, Jamel S, Thomas SA. 2010. Central nervous system (CNS) delivery of glucocorticoids is fine-tuned by saturable transporters at the blood-CNS barriers and nonbarrier regions. Endocrinology 151:5294–5305. The statement should be revised.

Line 233-34. The authors describe a causal relation based on correlational data. Although they state this causation tentatively and cite several papers in support, they should also describe the evidence that informs the directionality of this claim. Although the reader could read the papers cited, its seems better to include a brief statement as the bases for the directionality claim.

Line 263. I believe it should be shown not showed.

Line 305. The word ‘of’ is needed between terms and cellular.

Line 371. The word ‘the’ is needed after into.

Line 432. The I in ‘in’ should not be capitalized.

Line 512. The I in ‘in’ should not be capitalized.

Line 541. An open bracket is missing.

Line 591. The font size changes.

Author Response

Thank you for this careful review. I have made the typographical corrections you suggest. I have also added a sentences to clarify that I was referring to the multidrug resitance protein (line 233-04), but D1 (lines 432, 512) remains - it is number one, not a capital letter, which is standard nomeclature for the type 1 receptor.

All corrections have been highlighted in yellow

Reviewer 2 Report

This is an excellent review. It is incredibly well written and the topic is highly relevant.

Author Response

Thank you for your kind comments